# SPoC: Search-based Pseudocode to Code

**Sumith Kulal,** * **Panupong Pasupat,** * **Kartik Chandra, Mina Lee,**
**Oded Padon, Alex Aiken, Percy Liang**
Department of Computer Science
Stanford University
{sumith,ppasupat,kach,minalee,padon,aaiken,pliang}@cs.stanford.edu

## Abstract

We consider the task of mapping pseudocode to executable code, assuming a one-to-one correspondence between lines of pseudocode and lines of code. Given test cases as a mechanism to validate programs, we search over the space of possible translations of the pseudocode to find a program that compiles and passes the test cases. While performing a best-first search, compilation errors constitute 88.7% of program failures. To better guide this search, we learn to predict the line of the program responsible for the failure and focus search over alternative translations of the pseudocode for that line. For evaluation, we collected the SPoC dataset (Search-based Pseudocode to Code) containing 18,356 C++ programs with human-authored pseudocode and test cases. Under a budget of 100 program compilations, performing search improves the synthesis success rate over using the top-one translation of the pseudocode from 25.6% to 44.7%.

## 1 Introduction

We consider the task of mapping natural language pseudocode to functionally correct computer programs that are long enough to have significant intermediate state (e.g., 10–20 lines) and perform non-trivial computations. Previous work on executable semantic parsing mainly focuses on translating short text descriptions to one-line programs [57, 47, 58, 59, 29, 11], and while recent work explored generating longer programs from text descriptions [30, 54, 39, 18, 19, 17], these programs are mostly evaluated on syntactic metrics (e.g., exact match and BLEU score) rather than functional correctness. In contrast, the program synthesis community emphasizes functional correctness, typically captured by a set of input-output test cases that the program must compute correctly [14, 13]. However, input-output pairs give no information about the intermediate states of the program, making it difficult to synthesize long programs.

Synthesizing a general class of programs of significant length and internal complexity is too challenging without some description of the steps of computation. To that end, we propose synthesizing programs from natural language *pseudocode* and *test cases*. The test cases provide the functional specification, while the pseudocode provides guidance for the intermediate computations the program should perform.

To synthesize a functionally correct program, instead of relying on the top-one translation of the pseudocode, we *search* over the space of possible translations to find one that passes the test cases. In this work, we view the desired program as a composition of segments of code, each aligned to a segment of natural language in the pseudocode. Figure 1 instantiates our setup: each pseudocode line translates to a line of code with approximately one or two atomic statements. Unlike treating the whole program as a single big chunk (too coarse) or decomposing into individual tokens (too fine-grained), semantically coherent segments of code (rendered as single lines here) form a natural

---

| $i$ | $x_i$ | $y_i$ |
|---|---|---|
| 1 | in function main | `int main() {` |
| 2 | let n be integer | `int n;` |
| 3 | read n | `cin >> n;` |
| 4 | let A be vector of integers | `vector<int> A;` |
| 5 | set size of A = n | `A.resize(n);` |
| 6 | read n elements into A | `for(int i = 0; i < A.size(); i++) cin >> A[i];` |
| 7 | for all elements in A | `for(int i = 0; i < A.size(); i++) {` |
| 8 | set min_i to i | `int min_i = i;` |
| 9 | for j = i + 1 to size of A exclusive | `for(int j = i+1; j < A.size(); j++) {` |
| 10 | set min_i to j if A[min_i] > A[j] | `if(A[min_i] > A[j]) { min_i = j; }` |
| 11 | swap A[i], A[min_i] | `swap(A[i], A[min_i]);` |
| 12 | print all elements of A | `for(int i=0; i<A.size(); i++) cout<<A[i]<<" ";` |
|   |   | `}` |

**Public test case 1 (out of 5):** 5 3 2 4 1 5 $\rightarrow$ 1 2 3 4 5
**Hidden test case 1 (out of 8):** 8 9 2 4 5 6 2 7 1 $\rightarrow$ 1 2 2 4 5 6 7 9

Figure 1: Given $L$ pseudocode lines $x_{1:L}$ (with indentation levels $\ell_{1:L}$) and public test cases, our task is to synthesize a program with code lines $y_{1:L}$. The program is evaluated against both public and hidden test cases.

abstraction that can still be described precisely and succinctly by natural language. Within this framework, a program can be synthesized by choosing a candidate translation for each pseudocode line. We can now search for a combination of code lines that form a program passing the test cases.

However, common search methods for language-to-code tasks (e.g., beam search [56]) only use the sparse reward of whether the program succeeds to navigate the search space. Without performing credit assignment to pinpoint the causes of program failures, it is difficult to guide search toward promising programs. Since 88.7% of failures during search are due to compilation errors, we propose to perform credit assignment based on information returned from the compiler: When a program fails to compile, we use *error localization* methods to identify which line of the program is responsible for the failure, and then focus the search on alternative translations of the pseudocode for that line.

We propose two error localization methods. The first uses a multiclass classifier to predict one of the code lines as the offending line, which is then down-weighted in subsequent search iterations. In contrast to previous error correction models [15], our model also uses the error message and pseudocode for prediction. This is crucial when the compilation error can be fixed in multiple ways, but only some of which are consistent with the pseudocode. The second method, prefix-based pruning, uses additional compilations to find a code prefix that causes the error. Unlike the classifier, the identified code prefix is guaranteed to be erroneous and can be blacklisted entirely.

For evaluation, we collected and release a new dataset, SPoC (Search-based Pseudocode to Code) containing 18,356 C++ programs (14.7 lines on average). In contrast to other language-to-code datasets [30, 36, 19], all programs contain multiple test cases for validation. In contrast to the closely-related NAPS dataset [56], which also contains test cases but only 6% human-authored pseudocode, all programs in SPoC have human-authored pseudocode of a consistent annotation granularity. Section 3 details the comparison between SPoC and related datasets.

Using the top-one translation of the pseudocode yields a success rate of 24.6% on the test set. Under a limited budget of 100 synthesis trials (i.e., 100 code compilations and executions), our best method achieves a success rate of 44.7%. The multiclass error localization model reduces the number of synthesis trials needed in 15.5% of the programs, with a median absolute reduction of 26 trials and a median relative reduction of 42%. On the other hand, prefix-based pruning slightly increases the number of compilations for easier problems, but is more effective on harder programs, making it outperform the multiclass classifier under larger budgets.

## 2  Problem statement

Figure 1 illustrates the setup of our synthesis task. Given (a) a sequence $x$ of $L$ *pseudocode lines* $x_1, x_2, \ldots, x_L$ and (b) $k$ *public test cases* in the form of input-output string pairs

$(T_1^{\text{in}}, T_1^{\text{out}}), \ldots, (T_k^{\text{in}}, T_k^{\text{out}})$, the task is to synthesize a program $y$ consisting of $L$ *code lines* $y_1, y_2, \ldots, y_L$. Each code line $y_i$ is a fragment of code whose semantics is described by the pseudocode line $x_i$. To simplify the task, the indentation level $\ell_i$ of each line is also given; we leave the process of predicting $\ell_i$ from the input to future work.

The synthesized program is *accepted* if it successfully compiles and passes all public test cases (i.e., the compiled executable prints the string $T_i^{\text{out}}$ after reading the input $T_i^{\text{in}}$) as well as $k'$ additional *hidden test cases* $(\tilde{T}_1^{\text{in}}, \tilde{T}_1^{\text{out}}), \ldots, (\tilde{T}_{k'}^{\text{in}}, \tilde{T}_{k'}^{\text{out}})$.

For training, we are given a set of examples where each example contains pseudocode $x$, a gold program $y$, and both public and hidden test cases. At test time, the system has access to pseudocode $x$, public test cases (not hidden ones), and a computation budget. For a fair comparison under different computing environments, we use the number of *synthesis trials* as the budget, where in each trial, the system can issue a single call to the compiler and execute the compiled program on public test cases. The system must output a single final program, which will be validated on both public and hidden test cases.

## 3 Dataset

Recall that our main goal is to synthesize functionally correct programs of significant length and complexity. To this end, we argue that it is important to have both a description of the intermediate computation and a functional specification of the program. While semantic parsing datasets with shorter, query-like programs [57, 7, 33, 4, 59, 23, 44, 55] usually have an execution environment (e.g., a database) to validate the programs, Table 1 shows that most existing language-to-code datasets with longer programs [30, 36, 19] lack mechanisms to validate the correctness of programs. This inevitably leads previous work to resort to proxy metrics, such as exact match accuracy, BLEU score, and tree node F1 score, which only measure syntactic similarity rather than functional correctness [30, 54, 39, 18, 19, 17].

One notable exception and the inspiration for our work is the NAPS dataset [56] which contains both a description (pseudocode) and a functional specification (test cases) of competitive programming problems. However, most of their pseudocode in the training data (4,923,000 examples from 16,410 programs) is generated by heuristic rule-based templates, which is less realistic compared to the human-authored counterpart (2,231 examples from 582 programs). Furthermore, the descriptions suffer from inconsistent granularity: the artificial pseudocode is fine-grained (e.g., "increase var0 by 1") whereas the human-written pseudocode tends to be abstract (e.g., "compute factorial") as the annotators were encouraged to provide high-level descriptions. This discrepancy is reflected by the ratio of the length of pseudocode to that of code, which is 1:1.82 in the artificial dataset, and 1:3.26 in the human-authored dataset.

As no existing dataset contains both high-quality human-authored description with a consistent level of granularity and a mechanism to validate functional correctness, we created a new dataset called SPoC (Search-based Pseudocode to Code), which consists of programs, pseudocode, and test cases. The programs are non-trivial solutions to competitive programming problems, and each program is paired with public and hidden test cases. To control the quality and granularity of the pseudocode, instead of collecting free-form descriptions, we segmented the code in a consistent fashion and collected natural language pseudocode for each segment from curated crowdworkers.

### 3.1 Data collection

**Programs and test cases.**  Similar to the NAPS dataset [56], we scraped competitive programming *problems* and their test cases from `codeforces.com`. Each problem has multiple *programs* submitted by participants as solutions to the problem. We collected accepted C++ programs from problems marked as the easiest level based on their metric. Based on our pilot study, we filtered out programs with constructs that are difficult to consistently annotate with pseudocode (i.e., programs with `#define` macros, classes, structs, templates, switch statements, and mallocs).

Table 1: Datasets for natural language to code. In contrast to other datasets, our SPoC dataset contains human-authored pseudocode with a consistent granularity of description and test cases.

| | MTG [30] | HS [30] | DJANGO [36, 30] | CONCODE[1] [19] | NAPS[2] [56] | SPoC |
|---|---|---|---|---|---|---|
| Programming language | Java | Python | Python | Java | UAST | C++ |
| Number of programs (total) | 13,297 | 665 | 18,805 | 2,184,310 | 17,477 | 18,356 |
| Lines per program (average) | 30.4 | 7.7 | 1 | 4.4 | 21.7 | 14.7 |
| Type of natural language input | — card text — | | comment | documentation | — pseudocode — | |
| Additional input | — card metadata — | | - | class context | - | - |
| Granularity of text description | program (class) | program (class) | line | program (method) | varies | line |
| Fraction of human-annotated text | 100% | 100% | 100% | 100% | 6% | 100% |
| Number of annotators (total) | n/a | n/a | 1 | n/a | n/a | 59 |
| Test cases | ✗ | ✗ | ✗ | ✗ | ✓ | ✓ |
| Number of test cases (average) | - | - | - | - | 7.5 | 38.6 |

Table 2: Examples of complex code lines and pseudocode lines in the SPoC dataset.

| | |
|---|---|
| read n values into array a and array b | `for(int i = 0; i < n; i++) cin >> a[i] >> b[i];` |
| read n and m in a loop, printing n*m/2 and a new line on each iteration | `while (cin >> n >> m) cout << n * m / 2 << endl;` |
| print all elements of ans | `for (int i = 0; i < ans.size(); i++) cout << ans[i];` |
| change max to i if tree[i] > tree[max] or max otherwise | `max = tree[i] > tree[max] ? i : max;` |
| if m and n are odd | `if (m % 2 != 0 && n % 2 != 0)` |
| if a is a digit return 1 | `if (a >= '0' && a <= '9') return 1;` |
| add s to q (q is a set) | `q.insert(s);` |
| add ok to ans (ans is an integer) | `ans += ok;` |
| add element a to v (v is a vector) | `v.push_back(a);` |

**Decomposition.**  We decompose each program into code lines. To obtain slightly higher-level descriptions for common constructs, we group any block with only one statement with the preceding control statement (e.g., the one-line for loop "`for (int i = 0; i < n; i++) cin >> x[i];`" allows a high-level description "read n values into x").

**Pseudocode.**  We recruited 59 crowdworkers on Amazon Mechanical Turk to write pseudocode for each line of code. The workers can see the whole program and are encouraged to vary the sentence structure while still maintaining semantic correctness. To our pleasant surprise, we were able to recruit a set of workers (as opposed to curated specialists) capable of annotating C++ code via a qualification round in which we manually inspected their annotations.

**Statistics.**  Our dataset contains 18,356 programs submitted for 677 programming problems. Each problem has roughly 27 programs, which are likely to have similar semantics yet different code syntax. Excluding closing braces and the common "`int main()`" line, each program contains an average of 14.7 lines (with the minimum of 1 and maximum of 457 lines of code). The average length of code lines is 9.08 tokens, while the average length of pseudocode lines is 7.86 tokens.

While many code lines in the dataset are simple statements such as assignment ("`i++;`") or input reading ("`cin >> n;`"), a significant number of pseudocode lines are non-trivial to translate. As illustrated in Table 2, some code lines contain multiple atomic statements, in which case the pseudocode tends to become more higher-level. Even single-statement lines may require understanding complex sentence structures, idiomatic expressions, or the context from other parts of the program.

**Training and test sets.**  To evaluate the generalization on unseen problems and annotation styles, we created two test sets. We generated the first test set TESTP by splitting based on problems: we held out 158 problems (23% out of 677 problems), which is equivalent to 1,820 programs (10.1% of all programs). The second test set TESTW is split by workers: we held out 7 workers (12% out

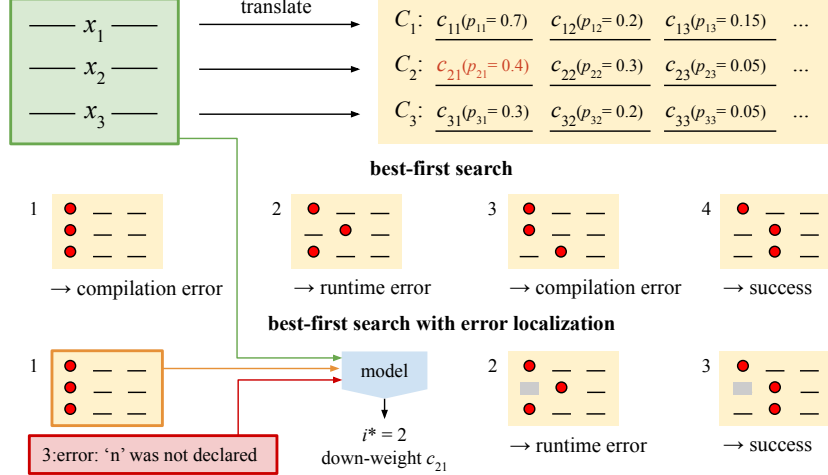

Figure 2: Illustration of best-first search and error localization model. In this example, $(c_{11}, c_{22}, c_{32})$ satisfies the test cases. Best-first search iterates in the order of decreasing probabilities and succeeds in four compiler calls. The error localization method down-weights $c_{21}$, leading to an earlier success.

of 59 workers), which is equivalent to 1,752 programs (9.7% of all programs, with 186 programs overlapping with TESTP). We used the remaining data for training and development (90:10 split).

## 4 Basic Approach

As illustrated in Figure 2, our starting point to synthesizing a program $y_{1:L}$ from pseudocode $x_{1:L}$ and public test cases involves two steps. First, a translation model takes each pseudocode line $x_i$ as input and generates $M$ candidate code lines $c_{i1}, \ldots, c_{iM}$ to be used as the $i$th code line. Then, we search over the possible combinations of candidate translations until we find a program $\hat{y}_{1:L}$ that successfully compiles and passes all public test cases.

**Translation.** To generate candidate code lines, we use the standard seq2seq implementation from OpenNMT [24] with an LSTM encoder and decoder, attention-based copying mechanism [32, 49], and coverage vector [46]. After encoding the pseudocode line $x_i$, we apply beam search with beam size $M$ to produce a ranked list of candidates translations $C_i = (c_{i1}, \ldots, c_{iM})$, where each code line $c_{ij}$ is a sequence of string tokens. (We use $M = 100$ for our experiments.) The model also assigns a probability $p_{ij} = p(c_{ij} \mid x_i)$ for each candidate $c_{ij}$. The translation model is trained on pairs $(x_i, y_i)$ from the training data using the standard log-likelihood objective.

**Best-first search.** Given the candidate lists $C_1, \ldots, C_L$, we can synthesize a program $\hat{y}$ by picking a candidate $c_{i,j[i]}$ from each $C_i$ (where $j[i] \in \{1, \ldots, M\}$) and then concatenating them into a program. In our search algorithm, we iterate through programs $\hat{y}$ in the descending order of probability $p(\hat{y}) = \prod_{i=1}^{L} p_{i,j[i]}$. To do so, we maintain a heap of the combinations $\hat{y} = (c_{1,j[1]}, \ldots, c_{L,j[L]})$ indexed by $p(\hat{y})$. The heap initially contains the program $(c_{11}, \ldots, c_{L1})$, which is the top-one translation of the pseudocode. In each iteration, we pop a program $(c_{1,j[1]}, \ldots, c_{L,j[L]})$ from the heap and test it. If the program fails (either from a compilation error, a runtime error, or a mismatch between the actual and expected test outputs), we push modified programs $(c_{1,j[1]}, \ldots, c_{i,j[i]+1}, \ldots, c_{L,j[L]})$ for all $i \in \{1, \ldots, L\}$ that have not been explored to the heap. We continue searching until we either find a program that passes all *public* test cases or exhaust the computation budget.

## 5 Error localization

So far, we have treated program compilation and execution as a black box that only tells whether a program passes the public test cases. This sparse signal offers little guidance: For instance, best-first search will keep using an erroneous candidate $c_{ij}$ if its probability $p_{ij}$ is high.

To speed up search, we unpack the black box. In this work, we focus on compilation errors, which constitute 88.7% of the failure cases in best-first search. When a program $\hat{y} = (c_{1,j[1]}, \ldots, c_{L,j[L]})$ fails to compile, the compiler reports error messages with associated line numbers. Unfortunately, the reported line numbers do not always correspond to the actual location of the mistake (e.g., the error "'*n' was not declared in this scope*" can occur long after the line where n should be declared according to the pseudocode). Empirically, the reported line number does not match the actual incorrect line 21.7% of the time.

Therefore, we treat the compilation error message as a noisy signal, and propose to use an *error localization method* to infer the actual portion of the code that causes the error. As illustrated in Figure 2, the error localization method has access to the pseudocode $x$, the synthesized code $\hat{y}$, and the *first* error message $(i_{\mathrm{err}}, m_{\mathrm{err}})$ from the compiler, where $i_{\mathrm{err}}$ is a line number and $m_{\mathrm{err}}$ is a message string. It then predicts an offending code line or abstains. We then either down-weight or blacklist the translation candidates in the offending code lines.

We now introduce two error localization methods: multiclass classification, which uses a neural model to predict a single offending line; and prefix-based pruning, which uses additional calls to the compiler for detecting an erroneous code prefix.

**Multiclass classification.** We train a classifier to predict the offending line $i^*$ among the $L$ lines. Our model is similar to the error correction model in [15]. For each line $i$, we embed the tokens of $x_i$, $y_i$, and $m_{\mathrm{err}}$, and then use three separate LSTMs to encode the sequences. We concatenate the final LSTM hidden states with the positional encoding [48] of the line offset $\Delta i = i_{\mathrm{err}} - i$, and then apply a feedforward network to produce the *line embedding* of line $i$. The $L$ line embeddings are then passed through another LSTM, and the hidden state of the LSTM cell corresponding to line $i$ is passed through a feedforward network to compute the logit for line $i$. We return the line $i^*$ with the highest probability (softmax over logits) if that probability exceeds a threshold $\beta_{\mathrm{mul}}$ and abstain otherwise. We use $\beta_{\mathrm{mul}} = 0.95$ for the experiments.[2]

Given $i^*$, we down-weight the current translation candidate of the line $i^*$ so that it is used less often in subsequent search iterations. Concretely, we multiply the probability $p_{i^*,j[i^*]}$ of the current candidate $c_{i^*,j[i^*]}$ in line $i^*$ with a constant factor $\alpha < 1$. As this affects the heap indices, we rebuild the heap from scratch (which takes negligible time) and continue the search, skipping any program that has already been explored before the heap rebuild.

To construct a dataset for training the model, we consider each program $y = y_{1:L}$ in the synthesis training dataset, substitute a single line $y_{i^*}$ with a candidate $c_{i^*j} \in C_{i^*}$ generated from pseudocode line $x_{i^*}$, and then collect any modified program $y'$ that produces a compilation error with an error message $(i_{\mathrm{err}}, m_{\mathrm{err}})$. The model is trained to maximize the log-likelihood of the offending lines $i^*$.

**Prefix-based pruning.** The multiclass classification method does not guarantee that the predicted line $i^*$ is actually an offending line. Furthermore, a candidate code line might be offending in some contexts but not others (e.g., a variable re-declaration is no longer offending if the previous declaration no longer exists). To address these issues, we propose an alternative that uses additional compiler calls to find an offending *prefix* of the program. Concretely, when a compilation error occurs, we use the compiler to to find the minimum $i^*$ such that the prefix $(c_{1j[1]}, \ldots, c_{i^*j[i^*]})$, plus closing braces to complete the program, fails to compile. Since programs containing that prefix will also fail (with very rare exceptions), we can safely *blacklist* the prefix from future search iterations.

Each additional compiler call is counted as one trial toward the synthesis budget. To use our budget sparingly, we only test $i^* = i_{\mathrm{err}} - \Delta i$ where $\Delta i \in \{0, 1, 2\}$ corresponds to the three most frequent offsets. If we fail to find an offending prefix, we simply abstain and continue the search.

## 6 Experiments

Our main evaluation metric is *success rate at $B$*: the fraction of test examples where the system generates an accepted program under the budget of $B$ trials. For error localization methods, we also consider the reduction in the number of trials used compared to normal best-first search.

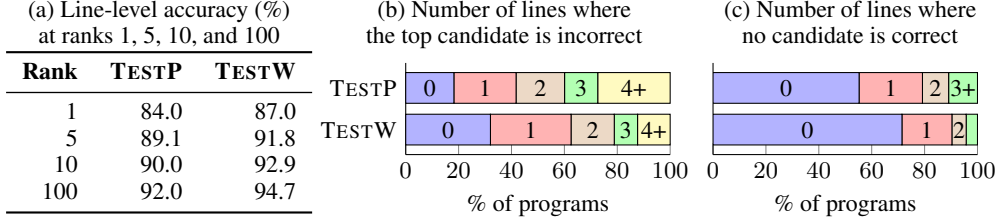

(a) Line-level accuracy (%) at ranks 1, 5, 10, and 100

| Rank | TESTP | TESTW |
|------|-------|-------|
| 1 | 84.0 | 87.0 |
| 5 | 89.1 | 91.8 |
| 10 | 90.0 | 92.9 |
| 100 | 92.0 | 94.7 |

(b) Number of lines where the top candidate is incorrect

(c) Number of lines where no candidate is correct

Figure 3: (a) While the translation accuracy is high at the line level, we need to consider the result at the program level. For each program, we count the number of lines $i$ where (b) the top candidate $c_{i1}$ is incorrect, and (c) none of the candidates $c_{ij} \in C_i$ is correct.

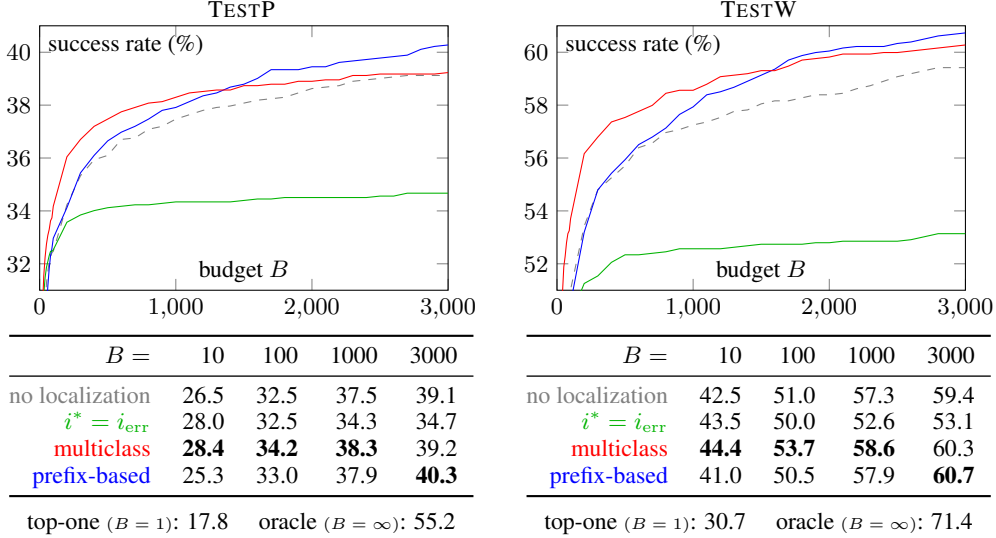

| $B =$ | 10 | 100 | 1000 | 3000 |
|-------|-----|------|------|------|
| no localization | 26.5 | 32.5 | 37.5 | 39.1 |
| $i^* = i_{\mathrm{err}}$ | 28.0 | 32.5 | 34.3 | 34.7 |
| multiclass | **28.4** | **34.2** | **38.3** | 39.2 |
| prefix-based | 25.3 | 33.0 | 37.9 | **40.3** |

top-one ($B = 1$): 17.8    oracle ($B = \infty$): 55.2

| $B =$ | 10 | 100 | 1000 | 3000 |
|-------|-----|------|------|------|
| no localization | 42.5 | 51.0 | 57.3 | 59.4 |
| $i^* = i_{\mathrm{err}}$ | 43.5 | 50.0 | 52.6 | 53.1 |
| multiclass | **44.4** | **53.7** | **58.6** | 60.3 |
| prefix-based | 41.0 | 50.5 | 57.9 | **60.7** |

top-one ($B = 1$): 30.7    oracle ($B = \infty$): 71.4

Figure 4: Success rates at budgets $B$ of best-first search with different error localization methods.

**Translation accuracy.** When evaluating the translation model, surface-form metrics such as exact sequence match and BLEU scores fail to account for functional correctness of the code. For instance, a prediction "`if (b)`" is functionally equivalent to the gold code "`if (b == true)`" when b is a boolean. Hence, we instead evaluate the *functional correctness* of the translation. To check if a predicted code line $c_{ij} \in C_i$ is functionally correct, we replace the code line $y_i$ in the gold program with $c_{ij}$ and then verify whether the program still passes both public and hidden test cases.

The results in Figure 3(a) shows that when the lines are considered independently, the translation model achieves a high accuracy of 84–87% under this notion of functional correctness. However, the picture is grimmer when we consider the statistics at the *program* level, which is what matters for synthesis. For each program, we count the number of lines $i$ where the top candidate $c_{i1}$ is not functionally correct. Figure 3(b) shows that only 18.2% of programs in TESTP and 32.0% of programs in TESTW have the top candidate correct in every line. As code lines that are functionally correct in isolation may be incompatible one another,[3] the programs formed by combining the top candidate of each line have an even lower success rates of 17.8% on TESTP and 30.7% on TESTW.

**Oracle success rate.** To compute the maximum achievable success rate given the lists of candidates, for each program, we count the number of lines $i$ where the candidate list $C_i$ does not have any correct candidate. Figure 3(c) shows that 44.8% of programs in TESTP and 28.6% of programs in TESTW have least one difficult line where the translation model does not produce a correct prediction among the top $M = 100$ candidates. This means a synthesizer with an infinite search budget would achieve a maximum success rate of 55.2% on TESTP and 71.4% on TESTW given our lists of candidates (assuming that incorrect candidates do not give a correct behavior when combined together).

Table 3: Effects of using error localization methods on all test examples.

| method | effect compared to best-first | number of trials: count | absolute difference mean | absolute difference median | relative difference geo.mean | relative difference median |
|---|---|---|---|---|---|---|
| multiclass | improves number of trials<br>failed to synthesize → succeeds | 13.5 %<br>2.0 % | −199.5 | −26.0 | ×0.39 | ×0.58 |
| | worsens number of trials<br>succeeded → fails to synthesize | 0.4 %<br>1.6 % | +407.5 | +123.0 | ×6.70 | ×7.04 |
| prefix-based | improves number of trials<br>failed to synthesize → succeeds | 4.1 %<br>1.5 % | −272.6 | −91.0 | ×0.45 | ×0.57 |
| | worsens number of trials<br>succeeded → fails to synthesize | 15.7 %<br>0.3 % | +68.4 | +12.0 | ×1.65 | ×1.63 |

```
(1)     …                          (2)     …
        let s be a string                  create int l, p and q
   7    string s ;                     2    int a , p , q ;
        read s                              read l, p and q
   8    cin >> s ;                      3    cin >> l >> p >> q ;
        if s is half                        print l * p / (p + q)
   9    if ( s / 2 == 0 )             4    cout << l * p / ( p + q ) << endl ;
        …                                  …
```

Figure 5: Examples of programs synthesized during search. In Program 1, prefix-based pruning detects that the prefix up to line 9 is offending. In Program 2, the multiclass model incorrectly predicts line 3 as the offending line, which ultimately leads to a failure.

**Synthesis results.**   Figure 4 compares the success rates of best-first search with and without error localization. As a baseline, we try down-weighting the reported error line ($i^* = i_{\mathrm{err}}$) whenever a compilation error occurs. Due to the mismatch between the actual offending line and the reported line, the synthesis result deteriorates. Up until the compilation budget of around $B = 1500$, the multiclass classifier improves the success rate more than prefix-based pruning. Prefix-based pruning achieves better success rates for higher budgets, but since it uses additional compilation calls, it performs worse under tighter budgets.

Table 3 details how the error localization methods affect the synthesis outcome. The multiclass model decreases the number of trials in 15.5% of all examples, but since its predictions are not verified, the model is also more prone to catastrophic failures. Prefix-based pruning uses additional compilation calls to verify its verdict, and thus slightly worsens the number of compilations needed in a large number of examples. However, for more difficult programs, the benefit outweighs the cost.

Between the two test sets, the success rate on TESTW is consistently higher than on TESTP for all methods. One possible explanation is that the unseen problems in TESTP require the model to potentially generalize to code constructs that are specific to solving those problems. As a piece of evidence, we found that 24.74% of the code token 4-grams in TESTP are unseen during training, while the number is only 15.77% for TESTW.

**Error analysis.**   To understand the behavior of error localization methods, we analyzed several examples from the development data. Some prototypical examples are shown in Figure 5. Program 1 shows how error localization can improve search. The condition "s is half" in line 9 should be translated as "`s == "half"`", but was instead interpreted as "`s / 2 == 0`" and "`s % 2 == 0`" with high probability, and hence best-first search spends a significant amount of budget (1511 trials) using these incorrect candidates in the combination. In contrast, prefix-based pruning detects them as offending candidates and succeeds earlier (413 trials).

In contrast, Program 2 shows how an incorrect error localization can lead to a catastrophic failure. Here, the multiclass model reads the error message $m_{\mathrm{err}} = $ "'l' was not declared in this scope" with line number $i_{\mathrm{err}} = 3$, and incorrectly predicts that line 3 is an offending line. This causes the search to ultimately fail whereas best-first search finds a correct program in 80 search iterations.

# 7 Related work and discussion

**Program synthesis.**  Program synthesis using test cases has been extensively studied in the programming languages literature. The most knowledge-heavy approach is to formulate synthesis as a constraint satisfaction problem [43, 45], which requires that the synthesis problem can be translated to a theory with effective constraint solvers. For other problems, brute force enumeration of programs (with some optimization) works surprisingly well [35, 3], but when the search space is too large for enumeration, randomized search guided by a cost function can be effective [41, 42]. Some works combine aspects of multiple approaches (e.g., [21]). For program specifications, the norm is to use input-output pairs. However, most synthesized programs are relatively short, and works that consistently synthesize longer programs are in the domains where the intermediate computation is easier to recover from input and output, such as string [14, 37, 8] and data structure transformations [13, 52]. For other domains, while input-output examples are *precise* in evaluating functional correctness, they provide mostly *global* signals and provide little information about the intermediate computations, thereby requiring other forms of specification along with input-output examples [12].

**Semantic parsing.**  Works on translating natural language specifications to executable programs, as discussed in Section 3, are closely related to semantic parsing whose goal is to map natural language utterances to formal representation. One of its traditional tasks is to parse a given question (usually a single sentence) into an executable database query [57, 58, 28, 4, 59, 53]. Instead of a single query, some work aims to parse a sequence of utterances into queries that can be sequentially executed [2, 20, 31]. However, the sequences are still relatively short (e.g., maximum 5 sentences).

While some semantic parsers operate by modifying the linguistic structure of the utterance [38, 40], most parsers construct the parse incrementally from scratch. Partial parses can be constructed by combining smaller partial parses in a bottom-up fashion [58, 28, 27], adding the next token to the sequence [10, 22, 51, 11, 25], or creating a new node in a tree structure [10, 26, 54, 39]. In any case, the search space of possible parses can be controlled by validating the constructed partial parses and pruning invalid ones [50, 6]. For instance, the parser can forbid partial parses that do not type-check, fail to execute, or execute into unfavorable values (e.g., an empty set). This validation process motivates our prefix pruning approach, which validates partial programs by invoking a compiler. However, we put an additional emphasis on minimizing the number of compilations needed, as our initial attempt to compile every prefix in the same fashion as previous semantic parsing works [50] significantly slows down the synthesis process.

**Error localization.**  Error localization in the context of automated program repair has been an active topic of research. Many recent work that uses neural models to localize errors has focused on localizing and correcting syntax errors [15] or a class of well-defined semantic errors such as variable misuse and variable replace [1, 9, 34]. Other work identifies error locations by interpreting compiler error messages [16, 5]. Likewise, our multiclass error localization model uses compilation errors to locate offending code; however, since the code is tied to pseudocode, we also use the signal from pseudocode to distinguish ambiguous cases (e.g., in Program 2 of Figure 5, while changing either line 2 or line 3 can fix the error, a correct model should choose line 2 as the offending line with respect to the pseudocode.)

**Acknowledgements.**  We thank Shivam Garg, Jason Koenig, Nadia Polikarpova, Alex Polozov and Rishabh Singh for valuable feedback at different stages of the project. This work was supported by NSF grant CCF-1409813, NSF CAREER Award IIS-1552635 and a grant from Amazon.

**Reproducibility.**  The dataset and code are available at `https://sumith1896.github.io/spoc/`. Reproducible experiments are available on the CodaLab platform at `https://worksheets.codalab.org/worksheets/0x23b27b2131634a158c8149d5b82adecf`.

## Footnotes

[1] We counted the number of programs in the released dataset. Since the programs are provided as a sequence of tokens, the number of lines per program is approximated based on the number of `;`, `{`, and `}`.

[2] We excluded partial programs (smaller pieces of full programs) in the dataset when counting.

[2] Refer to the Appendix for ablation studies of the design choices in the model.

[3] Functionally correct lines can also give an incorrect behavior when combined, but this occurs more rarely.

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
