[Supplementary Material · spoc_paper_appendix.pdf]

# Appendix: Search-based Pseudocode to Code

**Sumith Kulal,**[*] **Panupong Pasupat,**[*] **Kartik Chandra, Mina Lee,**
**Oded Padon, Alex Aiken, Percy Liang**
Department of Computer Science
Stanford University
{sumith,ppasupat,kach,minalee,padon,aaiken,pliang}@cs.stanford.edu

## A   Dataset samples - success

We present five samples from the dataset (dev set) that highlight good language features and were successfully synthesized. Corresponding synthesized code matches the gold code unless specified. Note that lines with ellipsis form a single line in the dataset.

| $i$ | $x_i$ | $y_i$ |
|---|---|---|
| 1 | in function main | `int main() {` |
| 2 | create int n | `  int n;` |
| 3 | read n | `  cin >> n;` |
| 4 | create vector of integers v with size n | `  vector<int> v(n);` |
| 5 | read n elements into v | `  for (int i = 0; i < n; i++) cin >> v[i];` |
| 6 | sort v | `  sort(v.begin(), v.end());` |
| 7 | for i from 0 to n - 1 exclusive | `  for (int i = 0; i < n - 1; i++) {` |
| 8 | if i is odd | `    if (i % 2 != 0) {` |
| 9 | remove first element from v | `      v.erase(v.begin());` |
| 10 | else | `    } else {` |
| 11 | remove the last element from v | `      v.pop_back();` |
|  |  | `    }` |
|  |  | `  }` |
| 12 | print the first element of v | `  cout << v[0] << endl;` |
|  |  | `}` |

Figure 1: Example 1 (success)

| $i$ | $x_i$ | $y_i$ |
|---|---|---|
| 1 | create long int array arr with size 1001 | `long int arr[1001];` |
| 2 | in function main | `int main() {` |
| 3 | create integers n, i and mid | `  int n , i , mid;` |
| 4 | read input to n | `  cin >> n;` |
| 5 | for i from 0 to n exclusive, read arr[i] | `  for (int i = 0; i < n; i++) cin >> arr[i];` |
| 6 | sort arr | `  sort(arr, arr + n);` |
| 7 | mid = n / 2 rounded up | `  mid = ceil(ceil(n) / 2 );` |
| 8 | print arr[mid - 1] | `  cout << arr[mid - 1] << endl;` |
|  |  | `}` |

Figure 2: Example 2 (success). Note: Gold code contains line 7 as `mid = ceil(double(n)/2);`.

[*]Equal contribution. The project webpage is at https://cs.stanford.edu/~sumith/spoc/.

| $i$ | $x_i$ | $y_i$ |
|---|---|---|
| 1 | create long long array num with 100010 elements | `long long num[100010];` |
| 2 | function cmp with integer arguments . . .<br>. . . a and b that returns boolean | `bool cmp(int a, int b) {` |
| 3 | return a > b | `  return a > b;`<br>`}` |
| 4 | in function main | `int main() {` |
| 5 | declare long long variable sum = 0 | `  long long sum = 0;` |
| 6 | declare integer n | `  int n;` |
| 7 | read n | `  cin >> n;` |
| 8 | for integer i = 0 to n exclusive | `  for (int i = 0; i < n; i++) {` |
| 9 | declare integer a | `    int a;` |
| 10 | read a | `    cin >> a;` |
| 11 | increase sum by a | `    sum += a;` |
| 12 | | `  }` |
| 13 | read n elements into num | `  for (int i = 0; i < n; i++) cin >> num[i];` |
| 14 | sort num using cmp as comparator function | `  sort(num, num + n, cmp);` |
| 15 | if num[0] + num[1] >= sum | `  if (num[0] + num[1] >= sum) {` |
| 16 | print "YES" | `    cout << "YES" << endl;` |
| 17 | else | `  } else {` |
| 18 | print "NO" | `    cout << "NO" << endl;`<br>`  }`<br>`}` |

Figure 3: Example 3 (success)

| $i$ | $x_i$ | $y_i$ |
|---|---|---|
| 1 | in function main | `int main() {` |
| 2 | create strings a and b | `  string a, b;` |
| 3 | read a and b | `  cin >> a >> b;` |
| 4 | if the length of a is not equal to the length of b | `  if(a.size() != b.size()) {` |
| 5 | print the max of length of a and length of b | `    cout << max(a.size(), b.size()) << endl;` |
| 6 | else | `  } else {` |
| 7 | if a is equal to b | `    if(a == b) {` |
| 8 | print -1 | `      cout << -1 << endl;` |
| 9 | else | `    } else {` |
| 10 | print the length of a | `      cout << a.length() << endl;`<br>`    }`<br>`  }` |
| 11 | return 0 | `  return 0;`<br>`}` |

Figure 4: Example 4 (success). Note: Gold code uses `x.length()` for length of x uniformly.

| $i$ | $x_i$ | $y_i$ |
|---|---|---|
| 1 | in function main | `int main() {` |
| 2 | create integers n, m, p | `  int n , m , p;` |
| 3 | get n, m, p | `  cin >> n >> m >> p;` |
| 4 | create integer ans with ans=0 | `  int ans = 0;` |
| 5 | create integer c with c=n | `  int c = n;` |
| 6 | while n not equal to 1 do the following | `  while(n != 1) {` |
| 7 | add n/2 to ans | `    ans += n / 2;` |
| 8 | subtract n/2 from n | `    n -= n / 2;`<br>`  }` |
| 9 | compute and print two expressions . . .<br>. . . ans*(m*2+1) and c*p . . .<br>. . . with a blank space between | `  cout << ans * (m * 2 + 1) << " " ...`<br>`    ... << c * p << endl;`<br>`}` |

Figure 5: Example 5 (success)

# B   Dataset samples - failure

We present four samples from the dataset (dev set) where all our synthesize techniques fail. First column contains the pseudocode, second column contains the top-1 prediction and third column contains the gold code.

| $i$ | $x_i$ | $y_i$ | $y_i^*$ |
|---|---|---|---|
| 1 | in function main | `int main() {` | `int main() {` |
| 2 | a = int array of length 1001, n = int, s = int = 0 | `int a[1001], n, s = 0;` | `int a[1001], n, sum = 0;` |
| 3 | read n | `cin >> n;` | `cin >> n;` |
| 4 | read n values into array a | `for (int i = 0; i < n; i++) { cin >> a[i]; }` | `for (int i = 0; i < n; i++) { cin >> a[i]; }` |
| 5 | for i = 1 to n inclusive | `for (int i = 1; i <= n; i++) {` | `for (int i = 1; i <= n; i++) {` |
| 6 | if a at i is 1 increment sum | `if (a[i] == 1) sum++;` | `if (a[i] == 1) sum++;` |
|  |  | `}` | `}` |
| 7 | print sum | `cout << sum << endl;` | `cout << sum << endl;` |
| 8 | for i = 1 to n inclusive | `for (int i = 1; i <= n; i++) {` | `for (int i = 1; i <= n; i++) {` |
| 9 | if a at i + 1 is 1 print a at i | `if (a[i] + 1 == 1) cout << a[i] << endl;` | `if (a[i + 1] == 1) cout << a[i] << " ";` |
|  |  | `}` | `}` |
| 10 | print a at n | `cout << a[n];` | `cout << a[n] << endl;` |
| 11 | return 0 | `return 0;` | `return 0;` |
|  |  | `}` | `}` |

Figure 6: Example 6 (failed). The model confuses the red part with `a[i] + 1`. Since this does not cause a compilation error, error localization methods are not of much help here. There is also a typo in the declaration of `sum` and the model cannot recover that.

| $i$ | $x_i$ | $y_i$ | $y_i^*$ |
|---|---|---|---|
| 1 | array of 1000 = a | `char 1000[1000] = {};` | `int a[10000];` |
| 2 | in function main | `int main() {` | `int main() {` |
| 3 | integers = n | `int n;` | `int n;` |
| 4 | read n | `cin >> n;` | `cin >> n;` |
| 5 | read in a until i = n inclusive | `for (int i = 1; i <= n; i++)` | `for (int i = 1; i <= n; i++) { cin >> a[i]; }` |
| 6 | for i = 0 to n inclusive | `for (i = 0; i <= n; i++) {` | `for (i = 0; i <= n; i++) {` |
| 7 | if a[i] modulo 2 is 0 then do… … the following a[i] -= 1 | `if (a[i] % 2 == 0) a[i] = 1;` | `if (a[i] % 2 == 0) a[i] -= 1;` |
| 8 | if i = n then do the following | `if (i == n) {` | `if (i == n) {` |
| 9 | output a[i] | `cout << a[i];` | `cout << a[i] << endl;` |
| 10 | else | `} else {` | `} else {` |
| 11 | output a[i], | `cout << a[i] << endl;` | `cout << a[i] << " ";` |
|  |  | `}` | `}` |
|  |  | `}` | `}` |
| 12 | return 0 | `return 0;` | `return 0;` |
|  |  | `}` | `}` |

Figure 7: Example 7 (failed). Line 1 does not specify a type, which leads to a large number of irrelevant candidates. Lines 9 and 11 are also ambiguous and cannot be detected by error localization methods. The beam search recovers rest of the bad lines (for e.g. line 5).

| $i$ | $x_i$ | $y_i$ | $y_i^*$ |
|---|---|---|---|
| 1 | set N to 2e6 + 5 | `N = 2e6 + 5;` | `const int N = 2e6 + 5;` |
| 2 | create long long a[N] | `long long a[N];` | `long long a[N];` |
| 3 | in function main | `int main() {` | `int main() {` |
| 4 | create int n | `int n;` | `int n;` |
| 5 | read n | `cin >> n;` | `cin >> n;` |
| 6 | for i=0 to 2*n, read a[i] | `for (int i = 0; i < 2 * n; i++) cin >> a[i];` | `for (int i = 0; i < 2 * n; i++) cin >> a[i];` |
| 7 | sort elements between a and a + 2*n | `sort(a, a + 2 * n);` | `sort(a, a + 2 * n);` |
| 8 | set ans = (a[n - 1] - a[0]) * (a[2 * n - 1] - a[n]) | `ans = (a[n - 1] - a[0]) … …* (a[2 * n - 1] - a[n]);` | `long long ans = (a[n - 1] - a[0]) … …* (a[2 * n - 1] - a[n]);` |
| 9 | for i=1 to n exclusive, set and to… … min(ans, (a[i + n - 1] - a[i]) * (a[2 * n - 1] - a[0])) | `for (int i = 1; i < n; i++) ans = …min = min(ans, (a[i + n - 1] - a[i])… …* (a[2]) * (a[2]) * (a[0])));` | `for (int i = 1; i < n; i++) ans = …min(ans, (a[i + n - 1] - a[i])… …* (a[2 * n - 1] - a[0]));` |
| 10 | print ans | `cout << ans << endl;` | `cout << ans << endl;` |
| 11 | return 0 | `return 0;` | `return 0;` |
|  |  | `}` | `}` |

Figure 8: Example 8 (failed). Line 9 is complicated, the translator needs to copy a large expression but could not generate a good candidate for it. The variable declarations missing in line 1 and 8 are recovered later in the beam.

| $i$ | $x_i$ | $y_i$ | $y_i^*$ |
|---|---|---|---|
| 1 | in function main | `int main() {` | `int main() {` |
| 2 | create integer n | `  int n;` | `  int n;` |
| 3 | let s be an array of characters of length 100 | `  char s[100];` | `  char s[100];` |
| 4 | read n and s | `  cin >> n >> s;` | `  cin >> n >> s;` |
| 5 | t is integer = 1 | `  int t = 1;` | `  int t = 1;` |
| 6 | for i = 0 to n exclusive | `  for(int i = 0; i < n; i++)` | `  for(int i = 0; i < n; i += t)` |
| 7 | print s[i] | `    cout << s [ i ] ;` | `    cout << s [ i ] ;` |
| 8 | increase t by one | `    t++;` | `    t++;` |
| 9 | print new line | `  cout << endl;` | `  cout << endl;` |
|  |  | `}` | `}` |

Figure 9: Example 9 (failed). Typo in the pseudocode (not specifying the loop increment condition) and the translator is not able to recover from it.

## C  Precision-recall trade-off

Figure 10: (a) Precision and recall of error detection methods on the held-out error localization examples. A higher threshold $\beta_{\mathrm{mul}}$ gives higher precision and lower recall. (b) Success rates at 100 and 1000 on development data when the threshold of the error localization method vary.

From the dataset for training the multiclass classification model, we hold out 10,000 held-out examples $((x, y', i_{\mathrm{err}}, m_{\mathrm{err}}), i^*)$ with gold offending lines $i^*$. On these examples, we compare the precision (fraction of correct predictions over non-abstained examples) and recall (fraction of correct predictions over all examples) of the error detection methods. (For prefix-based pruning, we check if the blacklisted prefix includes the gold offending line.) Figure 10(a) shows the precision-recall trade-off for different threshold values $\beta_{\mathrm{mul}}$. Note that while prefix-based pruning achieves high precision and recall, it comes with a price of using more compilation calls.

To investigate how precision and recall affect the success rate, we apply the multiclass classifier with different thresholds $\beta_{\mathrm{mul}}$ on 314 development examples that are improvable by search (i.e., the top-one translation fails but a good combination of candidates exists). The resulting success rates at 100 and 1000 in Figure 10(b) shows that models with higher precision tend to perform better for a higher budget, but recall is also important for a lower budget. One possible explanation is that it is harmful to down-weight candidates that are actually correct. Moreover, offending lines that could not be detected in one context might be easy to detect in other contexts, so it is safer to abstain than to make an incorrect prediction (though abstaining too often can also slow down the search).

## D   Model ablation

Figure 11: Precision and recall of different variants of the multiclass error detection model. Different points on the plot correspond to different thresholds $\beta_{\mathrm{mul}}$.

We analyze the effects of each input embedding on the multiclass error detection model. We found that the embeddings of the code, pseudocode, error message, and error line (positional encoding) all contribute to the model, with error message and lines being very crucial for identifying the correct offending line.

We also try replacing the positional encoding of the error line offset with a simple binary encoding. Concretely, we use two trainable embeddings, one for the reported error line and one for all other lines. The resulting model performs similarly to the original model. This is perhaps because our final bidirectional LSTM, which runs over all lines, can propagate the position information from the error line to other lines, which mimics the behavior of positional encoding.