[Reviews · NeurIPS 2019]

Reviewer 1



= Originality Improving the translation procedure by leveraging error detection on the target programming language is not a new idea, even though the authors do not seem to be aware of this. In semantic parsing, this has been used quite a bit over the last year, under the name of "execution-guided decoding" (see for example the WikiSQL leaderboard, where the top 4 entries currently use this). I believe this first appeared in "Execution-Guided Neural Program Decoding" (Wang et al, 2018); the idea is closest to the prefix-based pruning idea in this submission. This leaves the new "multiclass classification" error localization technique as original contribution, as well as the new dataset. = Clarity The core contribution of the paper is a new algorithm, which is only described in prose; I would have preferred pseudo-code here, which would have been more concise. There are a number of technical choices that are not explained in detail, nor evaluated in the experimental section. For example, the positional encodings in the mutliclass classification error detection setting were surprising to me. I would speculate they may help because (a) the true error never being /after/ i_err (is that the case?), (b) positional encodings allowing to filter out later lines, (c) positional encodings allow to downweight lines that are far away from the reported error location. [The authors have provided some additional information on this in their response, and it would be nice to include this in the next iteration of the paper as well] This could be tested by an alternative version in which the positional encodings are replaced by a single bit for 1_{i_err = i}, and removing all lines after i_err from consideration. I would also be curious to see alternative sequence models here (e.g., a simple self-attention layer). The authors also never comment on the fact that the transfer from training to unseen programs seems to work significantly less well than the transfer from training workers (and idioms) to new workers. This seemed like a very surprising result to me, and merits some analysis. = Quality The presented decoding strategy seems sound, and the experiments seem to be carried out to a reasonable standard. = Significance While I agree with the authors' point on the problematic shortness of existing program synthesis tasks, I am not convinced that their pseudo-code to code scenario is realistic: In what setting can we expect a user to provide a line-by-line description of the target program? I cannot conceive of a user that would able to provide all the algorithmic detail, but not be able to write this in an actual programming language. The opportunity for machine learning is exactly in the resolution of ambiguity of high-level user descriptions, not in the mechanistic translation from statements like "set min_i to j if A[min_i] > A[j]" to code. Hence, I do not expect the new dataset to be of substantial impact. While it allows for evaluation of program decoding with very strong supervision, I do not see that as the crucial problem in the research area. [The authors also do not comment on their relationship to approaches that use knowledge about the target language's syntax and semantics in their decoder and subsequently lead to almost no compilation errors (e.g. "Generative Code Modeling with Graphs", Brockshmidt et al., 2019)]

Reviewer 2



==================== update after author response & discussion ==================== Thanks for your response to our reviews. I'm happy to recommend acceptance, assuming that the promised tweaks will be made and any comments from the Meta Reviewer taken into account. ==================== original review ==================== Originality: medium-high Quality: high Clarity: high Significance: medium-high Although the results are perhaps not astonishing, I think this paper explores an interesting direction, I enjoyed reading it and it made me think more about the problem. Questions & comments: - Is the pseudocode example in Figure 1 representative of the granularity of the pseudocode in the whole dataset? In the real-world this level of granularity might be arguably unnecessarily high -- I would at least expect lines 8-10 to be compressed into something like min_i = index of smallest element in A[i..n] On the other hand, line 94 says that "annotators were encouraged to provide high-level descriptions". Could you add a comment on (what you think) the impact of the pseudocode granularity would be on the method's performance? - (Line 32 and Related work) The idea to not just look at the top-1 prediction seems sensible, and using a neural component to guide a search procedure exploring more possible solutions until one consistent with an input-output specification is found, is an algorithm template that has been used in program synthesis before. I'm familiar with DeepCoder and PCCoder, but maybe you can find better examples. Coincidentally, DeepCoder also looks at a few (very easy) competitive programming problems. - (Lines 67-70) The requirement that the final source code have the same number of lines as the pseudocode seems artificial (real pseudocode may often require translating a single line into multiple lines of compilable code). Could you comment on the (lack of) importance of this restriction? Would anything break if more source code lines could be produced per pseudocode line, or does the method routinely sidestep this restiction by putting multiple semicolon-separated statements on a single line? - (Lines 95-96) What is the corresponding pseudocode:code length ratio for the SPoC dataset? Based on the information provided on line 122, would it be roughly 1:1.15 (assuming equal token lengths), which would make it lower than the NAPS synthetic dataset, or are the pseudocode tokens on average shorter than code tokens? - (Line 102) Could you please clarify what is meant by the sentence saying that consistent granularity of the descriptions is ensured "by design"? It seems that a non-automatic check might be needed to ensure that different codeworkers produce descriptions on an equal granularity level? The existence of the TestW test set seems to also suggest that different crowdworkers might produce systematically different descriptions. The next question is also related. - When annotating each line of code, are the crowdworkers allowed to take into account the semantics inferred from surrounding lines, or does the annotator have to translate each line of code into pseudocode independently, ignoring its context? For concreteness, given these two lines of code ... 7: prefix_sums[0] = a[0]; 8: for (int i = 1; i < n; i++) prefix_sums[i] = prefix_sums[i - 1] + a[i]; ... would it be valid to describe line 8 as "compute prefix sums of the array a into prefix_sums"? This interpretation relies on the fact that prefix_sums[0] was set to a[0] on the preceding line. Or would a more granular translation of line 8 into pseudocode be required? - Is there any reason except simplicity for the translation seq2seq model to be applied to each line independently, not conditioning on the context (surroundings) of each line of pseudocode, nor on the candidates generated so far? - (Line 178) Is the LSTM that the L line embeddings ae passed through bi-directional or uni-directional? Minor comments: [Line 212] "shows" -> "show" [Line 218] "one another" -> "with one another" [Line 219] "have an even" -> "have even" [Line 223] "have least" -> "have at least" [Line 226] And also assuming that it is always possible to combine individually correct candidates together to form a correct program. [Line 270] "Many" -> "Much"

Reviewer 3



The paper is clearly and well written. The proposed synthesis benchmark from natural language pseudo-code with I/O test cases opened new directions for tacking various challenges in program synthesis (e.g., credit assignment, execution-guided synthesis). However, an issue with the newly collected dataset is that it assumes line-by-line mapping between the pseudo-code and the machine-executable code. However, in real-world use cases it is common for pseudo code to "omits details that are essential for machine understanding of the algorithm, such as variable declarations, system-specific code and some subroutines."[1]. Have the authors considered relaxing this assumption of having line-by-line mapping between the pseudo-code and the C++ code? [1] https://en.wikipedia.org/wiki/Pseudocode **Update**: Thanks so much for the detailed response! To add a bit: (1) [Dataset] The newly proposed dataset considered multi-line code generation given manually annotated pseudo code. Compared to NAPS, it is a line-by-line annotation (finer granularity) with significantly larger amount of human-annotated examples. Compared to the other (relatively large scale) line-annotated Django dataset, examples is this dataset are at functional level (more context), with extra I/O examples to guide the search of possible programs. (2) [Synthesis Algorithm] The proposed synthesis method is a simple line-based approach without considering context, with error localization methods to prune the hypothesis space. For (1), the central argument is that the line-by-line annotation is far from real-world settings, where the alignment between user's intents and the target code blocks is latent. This assumption would make the proposed synthesis algorithm not applicable to such real world settings, neither would the proposed approach shed light on how to advance its modeling towards tackling this latent alignment problem. In contrast, existing benchmarks like NAPS consider a more realistic setting, although most inputs are synthetically generated. For (2), while the algorithm is inline with the general idea of execution-guided decoding, in my opinion, the idea of using a trainable error feedback mechanism in synthesizing open-domain multi-line programs still has novelty, compared with using a deterministic executor to guide the generation of domain-specific logical forms. To summarize, I feel the technical contribution from (1) might be limited apart from introducing a new dataset, while (2) would inspire future work on leveraging execution information for synthesizing complex programs. Therefore, I lowered my score to a weak acceptance.

[Author Response · NeurIPS 2019]

**Addressing R1, R2, and R3:** We thank all the reviewers for their valuable feedback. We will incorporate the suggestions and provide a more comprehensive discussion of related work in the final version of the paper.

The reviewers raised concerns that the line-by-line alignment between pseudocode and code in our setting is too restrictive. We want to first point out that our framework is much more general: we only assume that the desired program can be decomposed into blocks, each aligned to a segment of natural language. Unlike treating the whole program as one big chunk (too coarse) or decomposing into individual tokens (too fine-grained), semantically coherent code blocks form a natural abstraction that can still be described precisely and succinctly by natural language. To instantiate this framework, we created the SPoC dataset, in which most blocks are single lines for simplicity. However, some blocks contain multiple lines, in which case the description tends to become more higher-level. For example:

```
(a) read n values into array a and array b   →   for (int i = 0; i < n; i++) cin >> a[i] >> b[i];
(b) print all elements of ans                →   for (int i = 0; i < ans.size(); i++) cout << ans[i];
```

Moreover, even the translation of each statement is non-trivial. Many blocks in the dataset are of comparable complexity to many code generation and semantic parsing tasks, and several statements are context-dependent (e.g., (b) above requires knowing the type of `ans`). With these challenges, the dataset is already difficult, and we believe this is a good initial step to tackle the problem of combining natural language and test cases in a more complex setting than any previous work. Over time, we agree it would be good to increase the difficulty of datasets by increasing the block size and the vagueness of natural language. Finally, we believe even the current system could be useful in education (getting hold of syntax), programmer productivity (moving to a new language), accessibility (coding via speech), etc.

**Addressing R1's comments:** R1 noted that using error detection to improve generation is not a new idea. We are aware of incremental and execution-guided decoding, which are quite established in the semantic parsing literature, and we will add appropriate citations accordingly. Nonetheless, one distinction of our prefix pruning is the focus on minimizing the number of program compilations by selectively compiling prefixes that are likely to be erroneous. In our initial attempts where we compile every prefix in the same fashion as Wang18, the generation process slows down significantly. This prompted us to look beyond whether the prefix successfully compiles and use novel signals (e.g., compilation error message) to synthesize more programs with the given budget.

In the multiclass error detection model, while positional embedding allows us to softly rule out lines that are less likely as R1 suggested, our main rationale for positional embedding is that many types of errors typically occur at a specific offset from the offending line (e.g., "'else' without a previous 'if'" mostly occurs 2 lines after the actual offending line).

Transferring to unseen problems is arguably more difficult because the model has to potentially generalize to C++ structures or methods that are specific to those problems. We will add more analysis in the next version of the paper.

R1 argued that the ability to evaluate program decoding with strong supervision is not a crucial research idea. We first want to clarify that the test cases are not only used for supervision at training time; they are also fed as input to the system at test time to (1) allow the system to search for a correct program, and (2) evaluate the functional correctness of the generated program. For (1): having test cases at test time makes decoding a more challenging task than previous semantic parsing and language-to-code settings, as the system has to perform search over a combinatorially large space of code—the fundamental challenge of program synthesis. For (2): we want to emphasize the importance of functional correctness. Besides functional correctness being ultimately necessary for generating *working* programs, partial metrics like BLEU can be gamed by learning the code boilerplate, or by mapping parts without properly combining them.

Methods that use syntax and semantics to guide synthesis for the *whole* program (NB: most of our line translations are already syntactically correct) are not guaranteed to generate semantically correct programs or even compilable programs, as only some semantic information (e.g, variable types) can be tracked during decoding. To generate semantically correct programs, we still have to search over the space of possible programs, which can be complicated or inefficient under syntactic/semantic decoding constraints when the program is large, but is an interesting direction to explore.

**Addressing R2's comments:** Figure 1 is indeed representative of the granularity of pseudocode. We found that a coarser granularity leads to underspecified natural language descriptions (e.g., "run Euclid's algorithm on m and n").

The ratio of token counts between pseudocode and code is roughly 1:1.2 which is lower than NAPS synthetic data.

Regarding annotation: we asked crowdworkers to give one annotation for every block of code (single C++ statements and common compound statements) in the program, which ensures that the granularity of pseudocode annotations are fixed to be identical to how we separate the code into blocks. Additionally, the whole program is visible to the crowd workers and they are indeed allowed to take the surrounding context into consideration during the annotation process.

Since our emphasis is on the search techniques, we translate each line separately for simplicity, but a context-dependent translation system could be easily plugged in. We rely on search to eliminate candidates that do not fit in context.

[Meta-Review · NeurIPS 2019]

Main contributions: * New dataset of line-by-line, human-generated pseudocode for learning to map from descriptions to source code. * A new kind of execution-guided decoding method for leveraging error messages generated by the compiler while generating target code. The first stage generates a set of candidate translations from pseudocode to code for each line. The second stage enumerates over combinations of candidates, tries compiling them, and then learns to use the error messages to prioritize which combinations to explore next. There are three well-qualified reviewers who did a great job with their reviews and were active in the discussions. The discussions centered around the following points: * Is this dataset a step forward compared to NAPS? We do not think the answer is clearly yes. SPoC has more human-generated alignments, but the assumption of line-by-line alignment is stronger and less realistic than NAPS. We DO NOT want SPoC to be viewed as a replacement for NAPS, but rather an intermediate point between the synthetically generated strongly-aligned part of NAPS and the weakly-aligned human-generated NAPS annotations. * The proposed method relies strongly upon the strong line-by-line alignment, and in this sense it seems like a step backwards. However, we find the way of using the learned error localization to be a novel and interesting contribution in the context of execution-guided decoding (though methods like DeepFix also learn to localize and repair programs from error messages). * There is some concern about whether the technique would still be beneficial when using a stronger model, like even one that consumed all lines of pseudocode jointly, then produced the set of per-line candidates independently. It seems that the proposed decoder will help the most when there is ambiguity in the line-by-line translations, and some of this ambiguity would be resolved by conditioning on the full pseudocode. Overall: all three reviewers have reasonable positions, and this paper could go either way. However, AC recommends following the majority vote in this case and accepting.